# First ice thickness measurements in Tierra del Fuego at Glacier Schiaparelli, Chile

Guisella Gacitúa[1], Christoph Schneider[2], Jorge Arigony[3], Inti González[1,4], Ricardo Jaña[5], and Gino Casassa[1]

[1]Centro de Investigación Gaia Antártica, Universidad de Magallanes, Punta Arenas, Chile
[2]Geography Department, Humboldt-Universität zu Berlin, Germany
[3]Instituto de Oceanografia, Universidade Federal do Rio Grande, Rio Grande, Brazil
[4]CEQUA, Punta Arenas, Chile
[5]Instituto Antártico Chileno, Chile, Punta Arenas, Chile

**Correspondence:** Guisella Gacitúa (guisella.gacitua@gmail.com)

**Abstract.** Cordillera Darwin in Tierra del Fuego (Chile) remains one of the least studied glaciated regions in the world. However, this region being one of very few terrestrial sites at this latitude in the Southern Hemisphere has the potential to provide key information on the effect of climate variability and climate change on the cryosphere at sub-polar mid-latitudes of the Southern Hemisphere. Glacier Schiaparelli is located at the northern side of the Cordillera Darwin draining the north side of Monte Sarmiento (2187 m asl). Despite being one of the largest glaciers in the Cordillera Darwin no previous in situ observation of its ice thickness had been made neither at this glacier nor at any other location in the Cordillera Darwin. Ice thickness is one of the fundamental parameters to understand glaciers dynamics, constrain ice dynamical modelling and predict glacier evolution. In April 2016 we performed the first successful ice thickness measurements using terrestrial ground-penetrating radar in the ablation area of Glacier Schiaparelli (Gacitúa et al., 2020), https://doi.org/10.1594/PANGAEA.919331. The measurements were made along a transect line perpendicular to the ice flow. Results show a valley shaped bedrock with a maximum ice thickness of 324 m within a distinct glacier trough. The bedrock is located below current sea level for 51% of the transect measurements with a minimum of -158 m which illustrates that the local topography is subject to considerable glacier-related over-deepening.

## 1 Introduction

In recent decades, efforts have been made to improve the knowledge of the effects of climate variability on glaciers and associated ecosystems in the Southern Hemisphere's sub-polar region. However, an important part of this region, such as Cordillera Darwin in Tierra del Fuego, southernmost South America, remains poorly explored with critical gaps of information. Here we describe geophysical data from the first ice thickness observations in Cordillera Darwin. The study is part of an international multidisciplinary collaboration to decipher the impact of climate variability and climate change on the cryosphere in Patagonia and Tierra del Fuego (Meier et al., 2019, 2018; Malz et al., 2018; Weidemann et al., 2018b). The climate of this

region is characterised by the effect of year-round prevailing westerly winds, cool summer temperatures and high rainfall, particularly along the west side of the mountain regions (Garreaud et al., 2009).

Glacier retreat in the region has been occuring since the Little Ice Age (Davies and Glasser, 2012; Strelin et al., 2008). The spatial variability of glacier retreat within Patagonia and Tierra del Fuego responds to different ice dynamic processes given geographical and topographical conditions (Homlund and Fuenzalida, 1995; Porter and Santana, 2003). Yet most of the glaciers in the region are experiencing major mass loss in comparison with worldwide average rates (Pellicciotti et al., 2014). It has been concluded that the main cause of rapid retreat is the increase in mean annual temperatures (Rosenblüth et al., 1997; Villalba et al., 2003), although ice dynamics and topographic controls are also important (Porter and Santana, 2003).

Glacier Schiaparelli (24.78 km$^2$) (Bown et al., 2014) is the northernmost glacier of the Sarmiento Massif in the Western Tierra del Fuego (54° 23'S 70° 52'W) (Fig. 1). It flows towards NW, being exposed to atmospheric circulation from the Pacific Ocean being thus an indicator of glacial response to oceanic climatic variability related to both warming trends and variability in atmospheric circulation patterns (Weidemann et al., 2018a). The glacier calves into Lago Azul, which recent bathymetry observations (April 2018) show a lake depth of approximately 60 m at the calving front (subsequent publication data).

Since direct observations of glaciers, such as ice thickness, are crucial to understand ice dynamics, in April 2016 we carried out field work on Glacier Schiaparelli to obtain in situ data. Among other measurements, we collected the first set of ice thickness data of Glacier Schiaparelli (Gacitúa et al., 2020). This paper describes the methodology and results obtained using ground-penetrating radar. These data are valuable for further studies on ice dynamic modelling, climate impact research in Cordillera Darwin (Meier et al., 2019) and the evaluation of global ice thickness modelling (Huss and Farinotti, 2012; Farinotti et al., 2019).

## 2    Methodology

The ground-penetrating radar used (http://www.unmannedindustrial.com/sites/default/files/GPR.pdf) consists of an impulse system (transmitter/receiver), control unit (hand-held) and resistively loaded dipole antennas (8 m length each). The antennas length provides an approximate central frequency of 10 MHz, following the design by Wu and King (1965). The transmitter emits signals with voltage set to 1.4 kV in accordance with relatively shallow ice. The receiver amplifies and stacks the radar signal (traces), and communicates with the hand-held unit which controls the collection and stores the data. The system operates at 1 kHz PRF (pulse repetition frequency) and the trigger signal is conveniently synchronised by GPS devices incorporated at both transmitter and receiver providing position for each stored scan. The system can stack a maximum 4096 traces and captures maximum 4096 samples per scan. Final data is delivered as consecutive files of 1000 traces each. The receiver uses an analog-to-digital converter that operates at 80 MSPS (mega samples per second) and has a resolution of 16 bits. Both transmitter and receiver function with an external battery of 12 V and 7 Ah, allowing an autonomy of one day of measurements in ideal conditions.

The field operation requires three persons: the first holds one extreme of the receiver antenna, the second carries the receiver equipment and the third person carries the transmitter equipment. The collinear antennas are connected by the extremes with a rope of half dipole length, resulting in a 40 m total system length (see inset photo in Fig. 2).

## 3 Results

A nearly complete profile across the glacier of approximately 3.1 km (two-way transect) was performed on the lower part of Glacier Schiaparelli. Crevasses on the glacier prevented reaching the northern margin of the glacier. We adjusted the parameters, such as vertical range and resolution, during the data collection and obtained 7 files (varied size content). We assume a constant wave speed propagation of 0.168 m/ns in temperate ice (Johari and Charette, 1975). Data were processed using a commercial software (ReflexW, Sandmeier (2011)). The processing steps include a) geometric corrections of zero depth considering distance of 24 m between transmitter and receiver, b) bandpass frequency filter, c) re-sampling to correct different vertical resolution (number of samples per trace) of files, d) frequency-wave number (F-K) migration (Stolt, 1978) to reduce diffraction noise and correct the position of the bedrock reflectors and e) subtracting average values to reduce horizontal noise.

An estimation of depth-average attenuation rate was made based on the method described by Jacobel et al. (2009). We calculated the power reflected from the bedrock over one transect (A-B) and normalised it to eliminate the inverse square losses due to geometric spreading. We obtained a depth-average attenuation rate of 22.6 dB/km from the best fit line between the ice thickness and the normalised power reflected. Although radar-attenuation rates have been calculated in multiple studies in polar ice, these estimations have rarely been made for temperate glaciers. However, our result is in good agreement with those presented in MacGregor et al. (2015) for the Greenland Ice Sheet where they obtained up to 25 dB/km by the ice-sheet margin, where ice temperature/conditions might be similar to those of temperate ice in Cordillera Darwin.

Figure 3 shows the resulting compiled radar data with the manual picking of the bedrock interpretation. Ice depth is subject to at least 10% of error due to manual picking, geometrical variations, and the assumption of homogeneous ice. The radar data shows a steep U-shaped valley with a maximum ice thickness of 324 m within a distinct glacier trough. The data show that 51% of the bedrock is below current sea level (Fig. 4) reaching a minimum of -158 m within a distinct morphological over-deepening, presumably as a result of glacier erosion. An interpolation of the bedrock elevation was made using the glacier outline for the ablation area covered (Fig. 2). Despite the fact that the resulting interpolation suffers from the lack of bedrock data at the North and Northwest edges, it provides a fair representation of the valley shape. At the time of the measurements there were visible signs of a recent discharge of an ice-dammed lagoon (Fig. 2, upper right photo) located at the northern side of the glacier tongue. The mid-depth reflections close to B (Fig. 3) suggest we crossed on the surface what might have been a subglacial tunnel through which meltwater drained from the lagoon to the pro-glacial lake.

## 4 Conclusions

These first results provide a calculation of the ice thickness within the ablation area of Glacier Schiaparelli. Only airborne measurements could provide a full coverage of ice depth data of the glacier due to inaccessibility in crevassed areas. Further retreat of Glacier Schiaparelli will probably lead to an enlarged and strongly overdeepened Lago Azul proglacial lake.

5 *Data availability.* Dataset containing the georeferenced ice thickness measurements are available for further applications at https://doi.org/10.1594/PANGAEA.919331

*Author contributions.* TEXT

ChS, GC, RJ, JA and IG managed the general project under which GPR data were obtained. GG collected the data supported by the co-authors. IG and JA provided the bathymetry data and information. GG processed the GPR data and prepared the

10 manuscript with contributions from all authors.

*Competing interests.* We declare that no competing interests are present.

*Acknowledgements.* This research was funded by the CONICYT-BMBF project GABY-VASA (BMBF140052) "Responses of Glaciers, Biosphere and Hydrology to Climate Variability across the Southern Andes". The field campaign was sponsored by UMAG, INACH, CEQUA e INCT da Criósfera. We wish to thank Roberto Garrido, Stephanie Weidemann, Marcelo Arévalo and Valentina Peredo for their assistance

15 during field work. We thank the Chilean Navy for the logistic support.

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

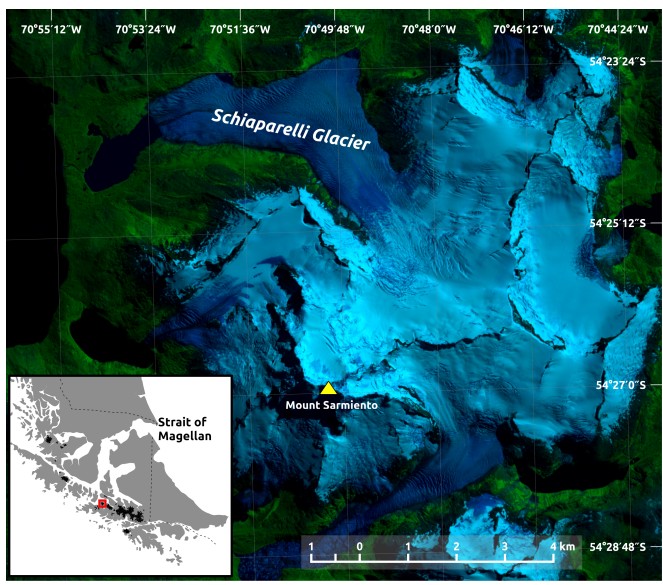

**Figure 1.** Glacier Schiaparelli location. The proglacial lake to the left (West) of Glacier Schiaparelli is named Lago Azul. The inset shows the southernmost tip of South America where Tierra del Fuego is south of the Strait of Magellan; the black areas depict the glaciated areas of Cordillera Darwin. *Image: Sentinel 2 RGB -84, 4th February 2019.*

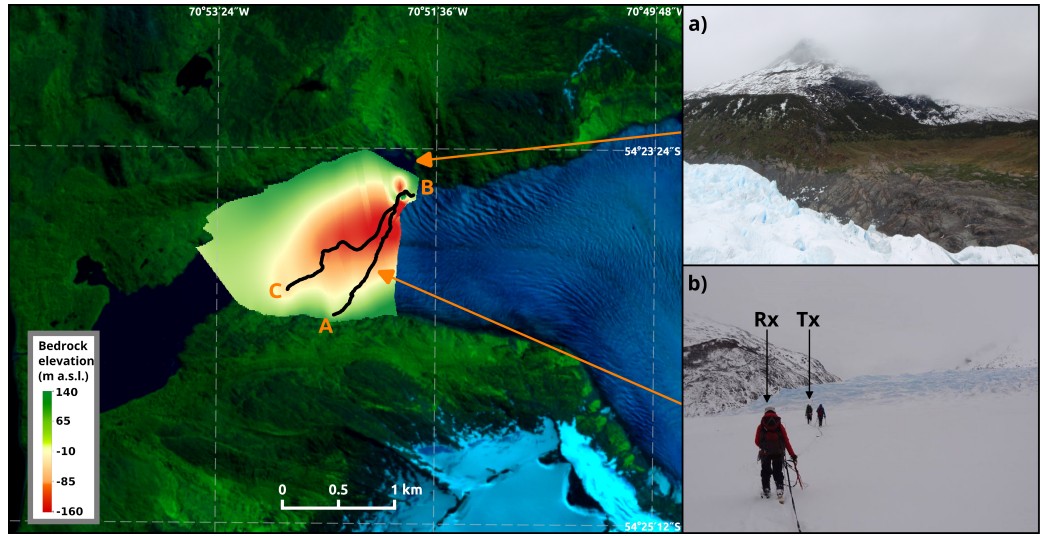

**Figure 2.** The radar track is shown in black. The coloured area depicts the interpolation of the bedrock elevation using the GPR data and surface data (SRTM, LP DACC NASA Version 3) as a reference, using Triangulated Irregular Network (TIN) grid. A, B and C indicate the respective positions in Figure 3. The upper frame (a) shows a photo taken during the GPR measurement from B towards the ice-dammed lagoon (north side). The lower frame (b) shows the team performing the measurements from A to B. Tx and Rx are the transmitter and receiver carriers respectively.

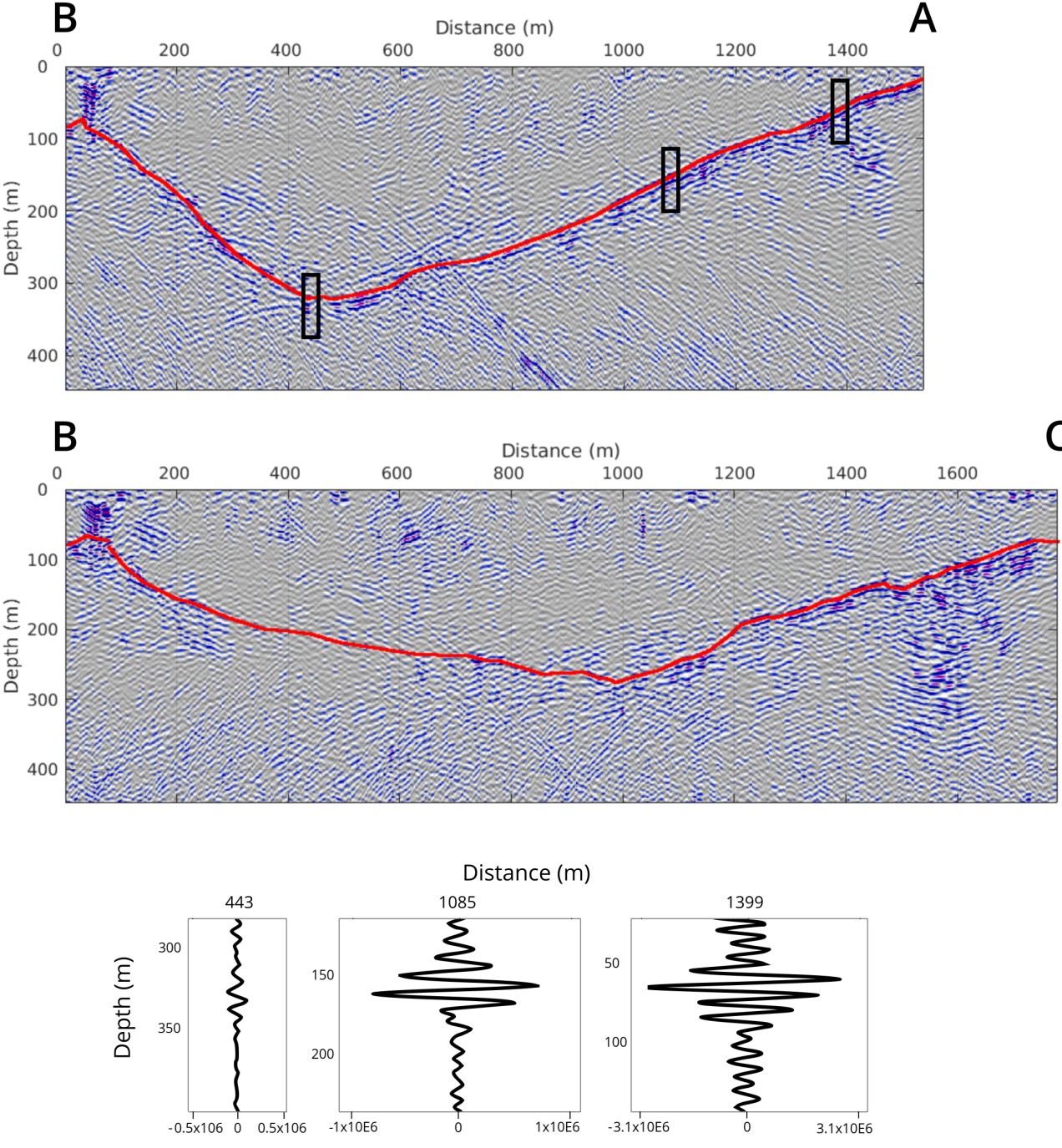

**Figure 3.** Resulting radargrams showing the manually interpreted bedrock in red. The x-axis shows the distance covered during the measurements. The y-axis represents the estimated bedrock depth in meters. The lower plot depicts A-scope examples of the attenuation of the signal at different depth from B to A.

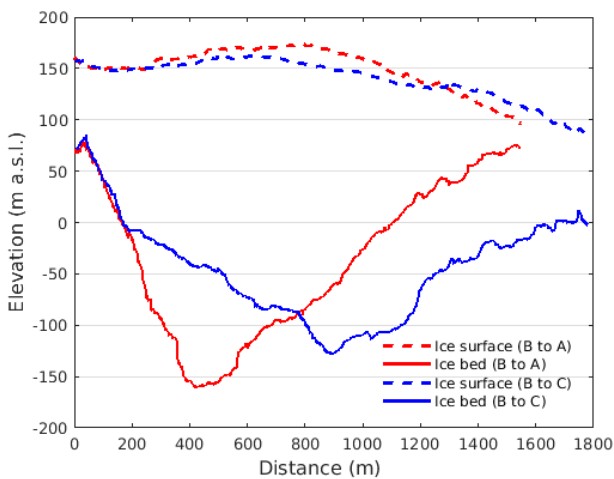

**Figure 4.** Ice surface and glacier bed elevation along the profiles shown in Figure 3.