# Peer review of "First ice thickness measurements in Tierra del Fuego at Glacier Schiaparelli, Chile"

_Earth System Science Data, 2020_

## Referee Comment (RC1) · Howard Conway (Referee) · 24 Aug 2020

The manuscript reports new radar-detected ice thickness measurements from Glacier Schiaparelli in Tierra del Fuego. These are important and fundamental measurements from a remote site on the west coast of South America (S54.38, W70.87); measurements from the region are sparse and difficult to obtain. The study is a contribution to an international collaboration to evaluate climate variability and climate change in Patagonia and Tierra del Fuego.

Ice thickness and surface elevation data from the study have been archived and are available at: https://doi.org/10.1594/PANGAEA.919331. This report provides details of the methods and results. The manuscript is well written and clear – nice work.

[Figure]

As an aside, I am interested to know whether the depth of proglacial Lago Azul been sounded? If not, might that be possible to do that from a small boat in the future?

A question about data processing: I am surprised that a bandpass filter to eliminate high- and low-frequency signals was not mentioned in the results section. I am speculating that a filter might help improve the resolution of the bed reflection, thereby reducing the uncertainty in the bed picks. Does the software package allow you to apply a bandpass filter and adjust the bandwidth of the filter?

And two spelling corrections - Line 19: "multidisciplinary" rather than "multidiciplinary" - Figure 2 Caption "Triangulated" rather than "Triagulated"
* * *

---

## Referee Comment (RC2) · Anonymous Referee #2 · 17 Sep 2020

General Comments: This manuscript reports the first ice thickness measurements on Schiaparelli glacier, Chile. The paper is well-written and presents a succinct account of recent in-situ measurements performed with a portable, commercial-grade ground penetrating radar system operating at a center frequency of 10 MHz. These are relevant and timely results because studies (and data) on the glaciers of Tierra del Fuego are limited. Glaciers in the Darwing mountain range are hypothesized to respond differently to climatic changes and thus ice thickness measurements such as these are needed to model and understand them better.

Specific comments: I only have the following small suggestions/corrections:

1) Please include a few "A-scope" plots (power vs. depth) for a few range lines shown in the echograms of Fig. 3. It would be helpful to do this for (1) shallow ice (<100 m)

[Figure]

as well as (2) the thickest ice sounded. Such plots will be helpful to estimate the ice attenuation and help guide the performance requirements for future radar surveys of these glaciers.

2) Please include an estimate of the ice losses in dB/km from the above. Comparisons with attenuations observed in other temperate ice glaciers should be included.

3) Small suggestion: Fig. 4, the ice bed profiles are displayed going from B to A (red trace) and then from B to C (blue trace). This helps making a comparison of the bed topography in the first 200 m (where the paths overlap). However, in Fig. 3, the echograms are shown going from A to B and then from C to B. I would recommend orienting the echograms in Fig. 3 to be consistent with the direction shown in Fig. 4.

4) Please confirm that the resolution of the ADC is 32 bits or otherwise clarify. Most commercial ADCs for ∼100 MSPS are 14-16 bits (that I am aware of). There are some 24-bit ADCs around, but they have lower sampling rates.

5) Page 3, line 7. There is a missing space between the number and the unit. It should read 24 m instead of 24m.

6) In Fig. 2(b), please mark the operators carrying the transmitter and receiver, respectively.

---

## Referee Comment (RC3) · Kenichi Matsuoka (Referee) · 13 Oct 2020

This is a concise manuscript reporting the first ice thickness measurement in one of the least known glaciated regions. It articulates the measurements and results well. I suggest following to increase clarity of the manuscript: 1. Legend in Fig. 2 says "depth (m a.s.l.)", which is confusing. I assume that it is "bed elevation (m a.s.l.)". 2. Figure 3 can be scaled for the horizontal distance.

Separately, the meta data at Pangaea needs a few more clarifications. 1. elevation (column 6), is this ice surface or bed surface? Clarify. 2. Latitude/longitude (columns 8 and 9), is this decimal degree? Clarify.

Similar measurements over the rest of the glacier using airplane and bathymetry mea-

surements in the proglacial lake are very useful and the former is mentioned by the authors. I hope such expanded work will be done in the near future to have a better understanding of this remote region in Southern America.

Reviewed by Kenny Matsuoka, Norwegian Polar Institute

―――――――――――――――――――

---

## Author Comment (AC1) · 11 Nov 2020

Response to reviewer 1

We thank reviewer Howard Conway for his evaluations and comments. We have addressed the concerns and made the suggested revisions to the text and figures. Below we show the reviewer comments in black, and our response in red.

The manuscript reports new radar-detected ice thickness measurements from Glacier Schiaparelli in Tierra del Fuego. These are important and fundamental measurements from a remote site on the west coast of South America (S54.38, W70.87); measurements from the region are sparse and difficult to obtain. The study is a contributionto an international collaboration to evaluate climate variability and climate change in Patagonia and Tierra del Fuego. Ice thickness and surface elevation data from the study have been archived and are available at: https://doi.org/10.1594/PANGAEA.919331. This report provides details of the methods and results. The manuscript is well written and clear – nice work.
As an aside, I am interested to know whether the depth of proglacial Lago Azul been sounded? If not, might that be possible to do that from a small boat in the future?

Yes, a group of scientists (co-authors of this paper) carried out bathymetry measurements at Lago Azul in April 2018. Preliminary results show a maximum depth of 60 m and the basin morphology reconstruction shows a U-shaped valley with transverse moraine ridges to the valley and glacier front.

Lines 12-13 were added in the Introduction to additionally describe Lago Azul. Further results are part of ongoing research and are material of a paper in preparation.

A question about data processing: I am surprised that a bandpass filter to eliminate high- and low-frequency signals was not mentioned in the results section. I am speculating that a filter might help improve the resolution of the bed reflection, thereby reducing the uncertainty in the bed picks. Does the software package allow you to apply a bandpass filter and adjust the bandwidth of the filter?

Yes, the software allows frequency filters. I originally eliminated low frequency noise for interpretation of the bedrock, which was made in full resolution of the radargram allowing a clear view of the reflector. However, we agree that eliminating high frequency noise improves the presentation of the image (lower resolution) in Figure 3, although it doesn't significantly improve the interpretation of the bedrock. We have added a bandpass filter to the processing flow and Figure 3 has been updated.

And two spelling corrections - Line 19: "multidisciplinary" rather than "multidiciplinary"- Figure 2 Caption "Triangulated" rather than "Triagulated"

These have been corrected.

---

## Author Comment (AC2) · 11 Nov 2020

Response to reviewer 2

We thank reviewer 2 for the comments and suggestions. We have addressed the suggested revisions to the text and figures. Below we show the reviewer comments in black, and our response in red.

General Comments: This manuscript reports the first ice thickness measurements on Schiaparelli glacier, Chile. The paper is well-written and presents a succinct account of recent in-situ measurements performed with a portable, commercial-grade ground penetrating radar system operating at a center frequency of 10 MHz. These are relevant and timely results because studies (and data) on the glaciers of Tierra del Fuego are limited. Glaciers in the Darwing mountain range are hypothesized to respond differently to climatic changes and thus ice thickness measurements such as these are needed to model and understand them better. Specific comments: I only have the following small suggestions/corrections:

1) Please include a few "A-scope" plots (power vs. depth) for a few range lines shown in the echograms of Fig. 3. It would be helpful to do this for (1) shallow ice (<100 m) as well as (2) the thickest ice sounded. Such plots will be helpful to estimate the ice attenuation and help guide the performance requirements for future radar surveys of these glaciers.

The presented radargrams are the resulting image of series of processes that include filters and gain adjustments. We agree that an estimation of the attenuation adds up to the work already presented, however the "A-scope" of the resulting processed image would not help to estimate the ice attenuation. We would prefer to keep this figure simplified to the raster and have added the numeric estimation of attenuation.

2) Please include an estimate of the ice losses in dB/km from the above. Comparisons with attenuations observed in other temperate ice glaciers should be included.

An estimation of depth-average attenuation rate was made based on the method described by Jacobel (2009) and a new paragraph was added in Results from Line 10 to Line 16.

3) Small suggestion: Fig. 4, the ice bed profiles are displayed going from B to A(red trace) and then from B to C (blue trace). This helps making a comparison of the bed topography in the first 200 m (where the paths overlap). However, in Fig. 3, the echograms are shown going from A to B and then from C to B. I would recommend orienting the echograms in Fig. 3 to be consistent with the direction shown in Fig. 4.

Revised. Figures are now consistent.

4) Please confirm that the resolution of the ADC is 32 bits or otherwise clarify. Most commercial ADCs for~100 MSPS are 14-16 bits (that I am aware of). There are some24-bit ADCs around, but they have lower sampling rates

Yes, we apologise for this error, the ADC is 16 bits and it has been corrected in the text.

5) Page 3, line 7. There is a missing space between the number and the unit. It should read 24 m instead of 24m.

Revised

6) In Fig. 2(b), please mark the operators carrying the transmitter and receiver, respectively.

Figure updated.

---

## Author Comment (AC3) · 11 Nov 2020

Response to reviewer 3

We thank Kenichi Matsuoka for his comments and suggestions. We have addressed the suggested revisions to the text and figures. Below we show the reviewer comments in black, and our response in red.

This is a concise manuscript reporting the first ice thickness measurement in one of the least known glaciated regions. It articulates the measurements and results well. I suggest following to increase clarity of the manuscript: 1. Legend in Fig. 2 says "depth(m a.s.l.)", which is confusing. I assume that it is "bed elevation (m a.s.l.)". 2. Figure 3 can be scaled for the horizontal distance.

Yes, that is correct. We have updated the text and figure.

Separately, the meta data at Pangaea needs a few more clarifications. 1. elevation (column 6), is this ice surface or bed surface? Clarify. 2. Latitude/longitude (columns 8and 9), is this decimal degree? Clarify.

Column 6 is Surface Elevation and Latitude/Longitude are in decimal degrees. We have requested to Pangea to update the header data.

Similar measurements over the rest of the glacier using airplane and bathymetry measurements in the proglacial lake are very useful and the former is mentioned by the authors. I hope such expanded work will be done in the near future to have a better understanding of this remote region in Southern America.

Airborne measurements are considered to be made. As mentioned in answers to reviewer 1, in 2018 a group of scientists (co-authors of this manuscript) carried out bathymetry measurements in the lake, a full description of these results are in preparation to be published.

---

## Author Response (AR2)

Response to the Topical Editor:

We thank the Editor for his comments and added the suggested A-scope examples at different depth in Figure 3.